# Protective Effects of Medicinal Plant-Based Foods against Diabetes: A Review on Pharmacology, Phytochemistry, and Molecular Mechanisms

**DOI:** 10.3390/nu15143266

**Published:** 2023-07-24

**Authors:** Prawej Ansari, Jannatul F. Samia, Joyeeta T. Khan, Musfiqur R. Rafi, Md. Sifat Rahman, Akib B. Rahman, Yasser H. A. Abdel-Wahab, Veronique Seidel

**Affiliations:** 1Department of Pharmacy, School of Pharmacy and Public Health, Independent University, Bangladesh (IUB), Dhaka 1229, Bangladesh; 2School of Biomedical Sciences, Ulster University, Coleraine BT52 1SA, UK; y.abdel-wahab@ulster.ac.uk; 3Natural Products Research Laboratory, Strathclyde Institute of Pharmacy and Biomedical Sciences, University of Strathclyde, Glasgow G4 0RE, UK; veronique.seidel@strath.ac.uk

**Keywords:** medicinal plant-based foods, pharmacology, diabetes, insulin, phytoconstituents

## Abstract

Diabetes mellitus (DM) comprises a range of metabolic disorders characterized by high blood glucose levels caused by defects in insulin release, insulin action, or both. DM is a widespread condition that affects a substantial portion of the global population, causing high morbidity and mortality rates. The prevalence of this major public health crisis is predicted to increase in the forthcoming years. Although several drugs are available to manage DM, these are associated with adverse side effects, which limits their use. In underdeveloped countries, where such drugs are often costly and not widely available, many people continue to rely on alternative traditional medicine, including medicinal plants. The latter serves as a source of primary healthcare and plant-based foods in many low- and middle-income countries. Interestingly, many of the phytochemicals they contain have been demonstrated to possess antidiabetic activity such as lowering blood glucose levels, stimulating insulin secretion, and alleviating diabetic complications. Therefore, such plants may provide protective effects that could be used in the management of DM. The purpose of this article was to review the medicinal plant-based foods traditionally used for the management of DM, including their therapeutic effects, pharmacologically active phytoconstituents, and antidiabetic mode of action at the molecular level. It also presents future avenues for research in this field.

## 1. Introduction

Diabetes mellitus (DM) is a severe medical condition that affects how the body processes glucose in the blood. It can develop following either a deficiency in the production of the anabolic hormone insulin or a lack of insulin sensitivity from cells or both. The resulting excess sugar in the blood leads to abnormalities in the metabolism of carbohydrates, lipids, and proteins. Insufficient production of insulin can also lead to these disruptions [1,2]. Diabetes is a widespread condition that affects a substantial portion of the global population, causing high morbidity and mortality rates and resulting in a major public health crisis. In 2021, it was estimated that over half a billion people have diabetes. This number is predicted to continue to rise, with 783.2 million people expected to live with diabetes by 2045 [3]. Diabetes is considered one of the five most severe diseases worldwide. The main symptoms of diabetes include increased blood glucose, excessive thirst, frequent urination, impaired vision, hyperphagia, weight loss, nausea, and vomiting [2,4].

Type 1 and Type 2 are the two most widespread forms of diabetes. Type 2 diabetes mellitus (T2DM) accounts for the majority of diabetes cases (approximately 90–95% of cases). Among type 1 diabetes patients, about 85% are reported with islet cell antibodies, which act against cells from the islets of Langerhans, affecting glutamic acid decarboxylase (GAD) function [2]. Type 1 diabetes (T1DM) is generally associated with insulin insufficiency due to the autoimmune destruction of pancreatic β-cells by CD4+ and CD8+ T cells and macrophages. The abnormal functioning of pancreatic ɑ-cells also contributes to the worsening of insulin insufficiency. The ɑ-cells produce an excessive amount of glucagon, which further leads to metabolic disorders. The decreased insulin and glucose metabolism in peripheral tissues contributes to raising the level of free fatty acids in the blood by triggering lipolysis. As a result, the target tissues fail to exhibit normal insulin responsiveness due to a deficiency in the glucokinase enzyme in the liver and glucose transporter (GLUT)-4 protein in adipose tissues [2]. On the other hand, the progression of T2DM is usually genetic and associated with obesity triggering a low capacity of β-cells to secrete insulin and insulin resistance [4]. In T2DM, chronic hyperglycemia is frequently observed in blood vessels of the cardiac, renal, and retinal circulation. The excess fat deposition in the blood vessels, heart, or peripheral tissues exacerbates insulin resistance and contributes to cardiovascular diseases [2]. Gestational diabetes is another type of diabetes that refers to glucose intolerance during pregnancy and heightened fetal–maternal complications [5]. Diabetes is interconnected to both micro and macrovascular complications, including retinopathy, nephropathy, neuropathy, ischemic heart disease, peripheral vascular disease, and cerebrovascular disease. This results in organ and tissue damage in one-third to one-half of all diabetic patients. The precise etiology of this damage remains elusive, although growing evidence suggests that oxidative stress and the generation of free radicals play a significant role [6].

Insulin therapy and several classes of antidiabetic drugs (i.e., biguanides, dipeptidyl peptidase-4 (DPP-4) inhibitors, meglitinides, sodium-glucose cotransporter-2 (SGLT2) inhibitors, sulfonylureas, and thiazolidinediones (TZDs) are currently available to treat diabetes and reduce the incidence of vascular complications [7,8,9]. However, many people in low- and middle-income countries find it difficult to obtain reasonably priced and widely accessible diabetes treatment options due to the unexpected rise in diabetes prevalence and associated medical expenses [3]. Another more holistic approach to the treatment of diabetes is the use of alternative medical systems such as Chinese Medicine, Unani, Ayurveda, and homeopathy [10]. Such traditional systems often use medicinal plants (e.g., turmeric, cardamom, garlic, onion, ginger, tulsi, and cloves) and other natural remedies that have become integral components of daily diets and are believed to possess antidiabetic properties [11]. Microorganisms have also presented a promising opportunity for the discovery of antidiabetic drugs. One such example is acarbose, a pseudo-oligosaccharide derived from various actinomycetes [11]. Recent studies have found that venom-derived compounds from cone snails, sea anemones, bees, scorpions, snakes, and spiders also possess effective antidiabetic properties [12]. The World Health Organization (WHO) has estimated that traditional medicine is used as a primary healthcare option by almost 80% of the global population. Over 800 plants have been reported to have antidiabetic properties, and these plants are often considered to have fewer side effects than synthetic drugs and contain bioactive compounds or phytochemicals with various biological properties [13,14,15]. The aim of this review is to discuss the medicinal plant-based foods traditionally used for the management of diabetes, including their active phytochemical constituents, therapeutic effects, molecular modes of action, and future prospects.

## 2. Methods

An extensive search was conducted using Google Scholar and PubMed databases to write this comprehensive review article. The keywords that were used included “Pathophysiology of diabetes”, “Prevalence of diabetes”, “Diabetes types and mechanisms”, “Diabetes epidemiology”, “Diabetes risk factors and management”, “Diabetes and Plant active components”, “Diabetes and plant dietary components”, “Antidiabetic properties of medicinal plants”, and others relevant to the topic. Over 400 articles were evaluated, and half of them were selected and incorporated into this article, ensuring only the most up to date, ranging from years 1998 to 2022, and relevant information was included.

## 3. The Pathophysiology of Diabetes

Diabetes mellitus is a dysregulation of glucose homeostasis either caused by the inability to produce insulin (in Type 1 diabetes) or an insufficient response to insulin (in Type 2 diabetes). DM occurs when the delicate balance of insulin and glucagon secretion in the pancreatic islets of Langerhans is disrupted due to alterations in the functioning of the insulin-producing β cells and glucagon-producing α cells [16]. Diabetes usually develops when fasting plasma glucose levels increase due to insulin resistance in the peripheral tissues (also known as the prediabetes stage). It further progresses to hyperinsulinemia, which is characterized by increased insulin production. The long-term overproduction of insulin causes cell failure, in turn contributing to hyperglycemia [17].

The increase in blood glucose levels beyond the normal physiological range in individuals with diabetes can lead to various complications, including renal, neural, ocular, and cardiovascular disorders, emphasizing the need for an early diagnosis of diabetes [17]. Polyuria serves as a crucial diagnostic hallmark for the early detection of diabetes and as an underlying factor in the pathogenesis of DM [18]. Hyperglycemia has also been shown to activate certain metabolic pathways, which contributes to the pathogenesis of diabetic complications [19]. Among these metabolic pathways, protein kinase C (PKC) activation plays a significant role in hyperglycemia-induced atherosclerosis. PKC activation is implicated in a variety of cellular responses, including growth factor expression, signaling pathway activation, and oxidative stress amplification. Hyperglycemia generally stimulates metabolic processes and increases ROS (reactive oxygen species) generation by activating the polyol and hexosamine pathways resulting in diabetes-induced atherosclerosis. The upregulation of the receptor for advanced glycation end product (RAGE) genes, which regulates cholesterol efflux, monocyte recruitment, macrophage infiltration, and lipid content in diabetic patients, triggers diabetes-induced inflammation [17]. Studies have shown that in diabetic mice, there is an increase in multiple PKC isoforms in the vasculatures of the renal glomeruli and retina. It has been observed that the activation of β- and δ-isoforms appears to be preferential. The activation of PKC in various tissues, including the retina, heart, and renal glomeruli, accompanied by the rise in blood glucose levels, exacerbates diabetic complications [20]. An increase in the total diacylglycerol (DAG) content has also been observed in various diabetic vascular complications, including in “insulin sensitive” tissues like the liver and skeletal muscles in diabetic animals and patients [21]. As a result of their production from glucose-derived dicarbonyl precursors, advanced glycation end products (AGEs) frequently accumulate intracellularly. AGEs are key triggers for the activation of intracellular signaling pathways and the alteration in protein activity [22]. Glycation disrupts the normal function of proteins by modifying their molecular shapes, affecting enzymatic activity, lowering breakdown capacity, and interfering with receptor recognition. Upon AGE degradation, highly reactive AGE intermediates (e.g., methylglyoxal, glyoxal) are formed. These reactive species are able to produce additional AGEs at a faster rate than glucose itself, fueling the production of AGEs and contributing to the pathogenesis of DM [23]. The pathogenesis of DM also involves the generation of pro-inflammatory mediators and ROS with elevated levels of cyclo-oxygenase (COX)-2, a crucial regulator in the conversion of arachidonic acid into prostaglandins that mediate inflammation, immunomodulation, apoptosis, and blood flow and elevated levels of antioxidant enzymes (glutathione S-transferase (GST), superoxide dismutase (SOD), and catalase (CAT)) counteracting the exacerbated oxidative stress [24].

T1DM is an autoimmune disease primarily seen in children and adolescents. T1DM is associated with the selective destruction of β cells, with no damage to other islets cells such as α cells (that secrete glucagon), δ cells (that secrete somatostatin), and pancreatic polypeptide cells (that modulate the rate of nutrient absorption). The development of T1DM is largely influenced by the rate of immune-mediated apoptosis of pancreatic β-cells. A strong connection has been established between damage to pancreatic β cells and genetics, as studies have revealed that variations in genes of the Human Leukocyte Antigen (HLA) complex increases T1DM susceptibility. In T1DM, mutations in such genes override self-tolerance mechanisms and result in the production of autoantibodies and T-cell cytotoxic to pancreatic β cells. This immune-mediated β-cell destruction and ultimate failure trigger diabetic ketoacidosis (DKA), typically considered the initial symptom of the disease. The presence of autoantibodies is an identifying trait of T1DM. These include autoantibodies to Glutamic Acid Decarboxylase (GADs) such as GAD65, autoantibodies to Tyrosine Phosphatases IA-2 and IA-2α, autoantibodies to the Islet-Specific Zinc Transporter Isoform 8 (ZnT8), Islet Cell Autoantibodies (ICAs) to β-cell cytoplasmic proteins like ICA512, and Insulin Autoantibodies (IAAs) [19,25].

T2DM is a metabolic disorder characterized by increased glucose levels, ROS generation, and inflammation, all of which are linked to obesity. The poor glycemic control in T2DM provokes ROS generation resulting in the stimulation of the redox pathway. Antioxidant enzymes (e.g., SOD, CAT, and GST), as well as COX, are produced. In T2DM, the β Langerhans cells are hypersensitive to glucose in blood plasma. As a result, they produce higher than normal insulin levels. This hyperinsulinemia serves to counteract hyperglycemia, which impairs β cell functions. Chronic hyperglycemia further induces microvascular complications resulting in higher morbidity and mortality [20]. Moreover, the accumulation of AGEs is a primary mediator in the progression of non-proliferative retinopathy in T2DM. The pathophysiological cascades triggered by AGEs also play a significant role in the development of diabetic complications. The accumulation of AGEs in the myocardium, observed in 50–60% of diabetic patients with microalbuminuria, has been linked to diastolic dysfunction and highlights the complex interplay between AGEs, oxidative stress, and diabetic complications [21]. Fatty liver, characterized by fat deposition in hepatocytes, is another key feature of T2DM. The high amounts of dietary lipids and abundance of free fatty acids from adipose tissues to the liver, as well as lipogenesis, are the main reasons for this metabolic imbalance [16]. Insulin resistance predominates in the liver and the muscles. The liver produces glucose from non-glucose substances (gluconeogenesis) in fasting periods to maintain a constant availability of carbohydrates. Increased gluconeogenesis is seen in hyperinsulinemia, suggesting that hepatic insulin resistance is an indicator of fasting hyperglycemia. The accumulation of fat in pancreatic islets ultimately contributes to β-cell dysfunction, leading to an increase in plasma glucose levels and a reduction in insulin response to ingested glucose [19].

Other types of diabetes include Maturity-Onset Diabetes of the Young (MODY) and gestational diabetes. In MODY, mutations occur in certain genes involved in insulin secretion by pancreatic β cells. This leads to a reduction in insulin secretion capacity. Gestational diabetes only occurs during pregnancy as a result of an increase in anti-insulin hormones, leading to insulin resistance and elevated blood sugar levels in the mother [5,9,19]. The presence of faulty insulin receptors can also result in a range of pathophysiological symptoms and complications associated with diabetes, including polydipsia, polyuria, weight loss due to calorie loss in urine, increased appetite (polyphagia), impaired wound healing, susceptibility to gum and other infections, cardiovascular disease, eye damage, kidney damage, nerve damage, and the risk of developing diabetic foot, diabetic ketoacidosis, and non-ketotic hyperosmolarity [9,26].

## 4. Medicinal Plant-Based Foods Recommended for the Treatment of DM

The management of DM is largely influenced by dietary habits and, in particular, the type of carbohydrates selected. The essential nutrients and phytochemicals found in whole grain-containing foods (e.g., brown rice, brown flour oatmeal, quinoa, millet or amaranth, roasted sweet potatoes), for example, have beneficial effects on hyperglycemia. Incorporating fiber-rich vegetables into the diet is also an effective way to manage blood glucose levels. Raw or lightly cooked vegetables, such as kale, spinach, and arugula, as well as frozen and canned (low-sodium or unsalted) vegetables, are all other good options for a diabetes-friendly diet. Fruits, on the other hand, provide a source of carbohydrates, vitamins, minerals, and dietary fibers, but contain a higher amount of carbohydrates than vegetables. It is recommended to choose fresh fruits, plain frozen fruit, canned fruits without added sugar, sugar-free or low-sugar jams or preserves, and no-sugar fruit-based sauces to maintain optimal diabetes control [27]. Black beans, tofu, tempeh, peas, lentils, common beans, garbanzo beans, green beans, and focus beans are varieties of canned or dried beans that are good choices for plant-based proteins and dietary fibers with potential antidiabetic effects. Studies have also suggested that the intake of dairy products, which are high in proteins, can have insulin secretory capabilities and help with T2DM. Parmesan cheese, ricotta cheese, cottage cheese, low-fat or skimmed milk, and low-fat Greek or plain yogurt are all excellent dairy additions to the diet. Moreover, proteins are generally high in fibers, low in fat, and take a long time to digest, generating relatively minor blood sugar spikes. Skinless boneless chicken breast or strips, salmon, sardines, tuna, other oily fish, white fish fillets, skinless turkey breasts, and eggs are also rich sources of proteins. The consumption of a diverse range of flavorings, seasonings, spices, herbs, and low-fat or sugar-free dressings is also beneficial to the control of diabetes, as these enhance the palatability of meals, promoting long-term compliance and ultimately better glycemic control. Vinegar, olive oil, mustard, any spice or herb, spicy sauces, and salsa are all good flavoring agents [28]. The section below discusses the antidiabetic effects of 20 common plant-based foods and their phytochemicals.

### 4.1. Althaea officinalis L.

*Althaea officinalis* L., commonly referred to as Marshmallow, Alcea, or Althaea, is a perennial species of the mallow family (Malvaceae). It is widely recognized for its medicinal properties and has been used for centuries in traditional Persian medicine, where it is referred to as Khatmi, Molukhia, or Panirak. In Western herbalism, it is known as Althaea, Moorish Mallow, Mortification Root, Schloss Tea, Sweet Weed, Mallards, Cheese, and White Maoow [29,30,31]. The plant is commonly found in marshy areas, especially near the sea as well as in damp areas throughout Western Asia and Europe. In North America, *Althaea officinalis* L. is naturalized in salt marshes ranging from Massachusetts to Virginia. The plant has since spread to other regions and is now cultivated in various parts of Western Europe and Russia [31].

All parts of marshmallows are used in cuisine, including their leaves, flowers, and roots [32]. The roots, leaves, and mucilage are used for medicinal purposes and contain many bioactive compounds such as flavonoids, coumarins, phenolic acids, and glycosides [29,33]. Marshmallow has been traditionally used for various diseases, including the management of diabetes and inflammation. Its use is widespread in Ayurvedic and Unani herbal medicine as ointments, suppositories, enemas, compresses, plasters, incense, and foot baths [29,31].

The plant contains pectin (11%), starch (25–35%), flavonoids (quercetin, kaempferol), mono- and disaccharide including sucrose (10%), coumarins (scopoletin), phenolic acids (protocatechuic acid, vanilic acid, chlorogenic acid, caffeic acid, and p-coumaric acid), tannins, phytosterols, amino acids, vitamins, and minerals. The seeds containing linoleic acid and mucilage rich in galacturonic rhamnans, arabans, glucans, and arabinogalactans can lower oxidative stress of clonal pancreatic β-cells and eventually alleviate diabetic inflammation [29,34].

Both in vitro and in vivo investigations of *A. officinalis* have demonstrated its strong potential for the treatment of DM. One study revealed that a marshmallow at a dose of 10, 30, and 100 mg/kg shows antihyperglycemic activity, reducing plasma glucose levels to 74%, 81%, and 65% of prior values measured before treatment in diabetic rats, respectively [31]. Another study reported that *A. officinalis* leaves (5% of the diet) significantly reduced cholesterol and serum glucose levels and markedly decreased glutamate pyruvate transaminase (GPT) in diabetic rats [33]. In addition, marshmallows, at doses of 250 mg/kg and 500 mg/kg, decreased blood glucose levels in streptozotocin (STZ)-induced diabetic animals compared to the control group [32]. Marshmallow has also demonstrated antioxidant activity, indicating its potential to reduce oxidative stress and alleviate diabetes [34]. Diverse phytochemicals such as flavonoids (quercetin, kaempferol), phenolic acids, tannins, and phytosterols are likely to contribute to the antidiabetic effects observed for this plant [29].

### 4.2. Anethum graveolens L.

*Anethum graveolens* L. from the Apiaceae (Umbelliferae) family is commonly known as ‘dill seeds’ or ‘dill weed’. This plant is found in Europe, throughout the Mediterranean region, as well as in central and southern Asia [35,36]. It is an annual or biennial aromatic herb widely cultivated across the world for commercial purposes [36]. Dill leaves are commonly used as food enhancers and flavoring agents. The leaves, stems, and seeds of this plant are traditionally used to treat digestive disorders, bad breath, and hyperlipidemia and to increase lactation and motivation [37,38].

Dill is rich in flavonoids, terpenoids, alkaloids, tannins, and phytosterols, which are responsible for the hypoglycemic effects of this plant. Alkaloids facilitate glucose transport to the peripheral tissues and stimulate insulin secretion, and saponin present in dill can inhibit AGE formation and heal diabetic neuropathy [38,39]. American dill has a high amount of α-phellandrene, while carvone and limonene are the main phytochemicals in Asian and European dill. The plant also contains lipids, proteins, carbohydrates, vitamin A, niacin, and various mineral elements (calcium, potassium, magnesium, phosphorus, and sodium) [39].

Dill has been reported to display a range of biological activities, including anticancer, antimicrobial, antiulcer, antioxidant, anti-inflammatory, and hypolipidemic properties [40]. It exerts its hypoglycemic effect mainly by interfering with antioxidant capacity and altering the expression of selected genes involved in the metabolism of glucose and lipids. Dill seed and leaf extracts, as well as its essential oil, have revealed hypolipidemic and hypoglycemic activities in diabetic mice, as evidenced by reduced triglycerides (TGs), total cholesterol, low-density lipoprotein cholesterol (LDL-C), very-low-density lipoprotein cholesterol (VLDL-C) and glucose levels, and increased high-density lipoprotein cholesterol (HDL-C) levels [39]. Studies carried out in animals and in humans have established that one month was sufficient to assess the hypolipidemic and antidiabetic properties of dill [39,41]. Dill increases LDL receptors in the liver and the uptake of LDL-C. It also inhibits the activity of Acetyl-CoA carboxylase and HMG CoA (3-hydroxy 3- methylglutaryl-coenzyme A) reductase and decreases fatty acid formation and cholesterol absorption from the intestine by binding to bile acids [39].

One study showed that dill markedly decreased AGE formation, protein glycation, and fructosamine levels and significantly reduced fasting blood glucose levels in diabetic animals [38]. Another study reported that the administration of an aqueous dill extract to diabetic rats possesses radical scavenging/antioxidant activity and significantly reduces fructosamine levels, protein carbonyl content, and thiol group oxidation, as well as β amyloid protein formation and fragmentation [39,40]. This study also found that after two months of treatment with dill extract on diabetic rats, there was a significant decrease in both blood glucose and AGE levels [40].

### 4.3. Allium sativum

*Allium sativum* (Alliaceae), more commonly known as Garlic, is one of the most cultivated and used herbs for both culinary and medicinal purposes [42,43]. The plant mainly originates from Central Asia and Iran and has been used in cooking and medicine since ancient times [44]. It is grown all over the world, but it was first cultivated in Asia before being naturalized in China, the Mediterranean region, Central and Southern Europe, Northern Africa, and Mexico [43,45].

Garlic is a flowering perennial plant that grows from a bulb. Its cloves are used for culinary purposes and possess nutritional benefits [46]. The Egyptians, Greeks, Babylonians, and Romans traditionally used garlic for its healing properties. In 1858, Pasteur discovered the antibacterial property of garlic, and it was later used during WWI and WWII as an antiseptic to prevent gangrene [45]. Garlic is used ethnomedicinally for a variety of purposes, including preventing infection, boosting immunity, and treating certain types of cancers, cardiovascular diseases, abdominal discomfort, diarrhea, respiratory tract infections, the common cold, asthma, hay fever, inflammation, fungal infections, hyperlipidemia, and diabetes [44,46].

Garlic contains 33 sulfur-containing compounds (mostly derived from the common precursor c-glutamylcysteine), 17 amino acids, various enzymes, and minerals such as selenium [45]. Sulfur compounds include ajoenes, thiosulfinates (allicin), vinyldithiins, sulfides, and alliin [47]. Additional constituents of intact garlic are lectins (the most abundant proteins in garlic), prostaglandins, fructan, pectin, adenosine, vitamins B1, B2, B6, C, and E, biotin, nicotinic acid, fatty acids, glycolipids, phospholipids, and essential amino acids [48]. Crushed garlic dry powder contains 1% of allin (S-allyl cysteine sulfoxide) and allicin (diallyl thiosulfonate) as their major constituents [45].

The sulfur-containing constituents of garlic are rapidly absorbed, transformed, and metabolized in the body [45,49]. Garlic has the ability to lower total cholesterol concentrations by approximately 10% and favorably alter HDL/LDL ratios. It can also lower blood pressure, enhance fibrinolytic activity, reduce blood clotting, and inhibit platelet aggregation. Moreover, it displays antioxidant and antidiabetic properties [45]. The antidiabetic effect of garlic is attributed to its ability to reduce blood glucose levels by decreasing the absorption of glucose from the intestinal tract. One study demonstrated that *A. sativum* increased the diameter of pancreatic islets of STZ-induced diabetic rats. This compound is a precursor of several other sulfur-containing compounds formed in aged or crushed garlic preparations. Allicin can enhance serum insulin activity due to its free SH group, which has antioxidative properties. It can also normalize oxidative stress and increase serum insulin levels in diabetic rats [43]. Aged garlic extract (AGE), containing compounds such as S-allylcysteine (SAC) and S-allylmercaptocysteine (SAMC), has been reported to possess even higher antioxidant and radical scavenging activity than a standard garlic extract. SAC has been reported to inhibit nitric oxide-induced intracellular oxidative stress in endothelial cells and reduce LDL oxidation and damage to endothelial cells caused by oxidized LDL. It can also scavenge intracellular peroxides and elevate intracellular GSH levels [49]. AGE has been demonstrated to decrease the risk of cardiovascular and cerebrovascular disease by inhibiting lipid peroxidation and LDL oxidation. Methiin, another sulfur-containing compound, and the flavonoid quercetin, both abundant in garlic, have been reported to reduce the severity of arteriosclerosis and lower serum cholesterol levels. Other compounds in garlic, such as steroidal saponins, have also been reported to reduce serum cholesterol concentrations [48]. Allicin is considered to be an effective insulin secretagogue as it binds with cysteine and protects insulin against SH group interactions, preventing insulin inactivation [45].

One study found that administering allicin at a dose of 200 mg/kg significantly improved DM by reducing blood sugar levels, which was comparable to the standard antidiabetic therapeutic agents glibenclamide and insulin. On the other hand, aged garlic extract (5 and 10 mL) prevented adrenal hypertrophy, hyperglycemia, and the rise of corticosterone on immobilization stress in diabetic mice. Another study documented that garlic oil (100 mg/kg) and diallyl trisulfide (40 mg/kg) administered over a period of 3 weeks to STZ-diabetic rats ameliorated basal insulin and oral glucose tolerance [43]. Garlic oil at a dose of 100, 250, and 500 mg/kg was also reported to significantly improve serum glucose, total cholesterol, TGs, urea, uric acid, creatinine, AST, and ALT in diabetic rats [49].

### 4.4. Brassica oleracea L.

*Brassica oleracea* L., also known as cabbage, is widely consumed across the world. Cabbage belongs to the Brassicaceae family, which includes other leafy vegetables such as broccoli, Brussels sprout, cauliflower, and kale. The Brassicaceae family has its origins in the Irano-Turanian Region and has since spread throughout the world. The entire cabbage plant, including its leafy flowers, is consumed either raw, cooked, or steamed. Cabbage is a shallow-rooted, cool-seasoned crop cultivated in varieties that include white cabbage, red cabbage, and savoy cabbage. It is a widely available and affordable vegetable with good nutritional value. White cabbage is particularly popular, as it is low in calories and rich in beneficial phytochemicals [50,51,52].

Before it was introduced as food, cabbage was initially used as a traditional medicine for headache, inflammation, digestion, various types of cancer, diarrhea, peptic ulcer, and gout as well as for detoxification purposes [51,52].

Cabbage is rich in fibers, minerals (Ca, P, and K), vitamins (A, K, C, tocopherols, folate) as well as diverse phytochemicals, including glucosinolates (sulfur and s-methyl cysteine sulfoxide derivatives), anthocyanins, coumarins, saponins, flavonoids, carotenoids, tannins, alkaloids, phenolic compounds, phytosterols, terpenes, and indoles [52,53].

In vitro studies have shown that cabbage can decrease postprandial hyperglycemia by inhibiting the activity of α-amylase and α-glucosidase, which are two enzymes responsible for the conversion of various oligosaccharides into free glucose after the consumption of carbohydrates [53]. In a recent clinical study, the effect of oral supplementation of red cabbage was evaluated on the glucose metabolism of STZ-induced diabetic rats. The results showed that the oral supplementation of red cabbage led to a statistically significant decrease in the levels of random blood glucose compared to the diabetic control group and to diabetic rats treated with Glibenclamide. Additionally, the levels of glycated hemoglobin (HbA1c) were found to be significantly reduced compared to the diabetic control group [54]. Cabbage has been demonstrated to control glucose homeostasis and improve hyperglycemic conditions in T2DM sufferers. It exerts multi-target effects on glucose homeostatic regulation and can mitigate organ damage in T2DM, particularly in the liver and kidney. It can also prevent problems underlying the development of T2DM, such as high oxidative stress and obesity [53]. It has been demonstrated that the regular consumption of white cabbage can lower LDL cholesterol and total serum cholesterol levels. White cabbage has free radical scavenging/antioxidant activity and inhibits lipid peroxidation [51].

Red cabbage is rich in anthocyanins. They are powerful antioxidant agents that have beneficial effects in controlling chronic diseases associated with oxidative stress, such as cardiovascular diseases, DM, and other metabolic disorders. Polyphenols, like anthocyanins, have been demonstrated to improve the structural and functional irregularities of pancreatic beta-cells in diabetes. Interestingly, other phenolic compounds such as flavonoids reduce platelet aggregation and vascular smooth muscle cell proliferation, thus proving of interest in the prevention and treatment of diabetic vascular problems [55]. Among the various vitamins found in cabbage, vitamin C has strong antioxidant properties. This vitamin can counteract the damaging effects of oxidative stress as well as reduce the production of pro-inflammatory cytokines, both involved in the pathogenesis of DM [50]. Vitamin K exerts its beneficial effect on DM by elevating insulin sensitivity [53]. The fibers found in cabbage, regardless of their type (soluble or insoluble), have been reported to ameliorate glucose control by reducing insulin resistance [54].

### 4.5. Cicer arietinum L.

*Cicer arietinum* (Chickpea) is one of the third most important and popular legume seeds worldwide and a good source of nutrients, especially proteins [56]. Chickpeas, also known as garbanzo beans, are a variant of pulses that are dry seeds belonging to the Leguminosae family [57]. Although the leaves and fruits can be employed, it is mostly the seeds of this plant that are used for consumption and medicinal purposes. Originating from Central and South America, the plant is now cultivated throughout the world [58]. India is currently regarded as the largest chickpea-cultivating country worldwide, contributing to about a 67% share of global production [57]. The seeds of chickpeas can be consumed either in their dried powder form or as cooked beans [56,59].

Chickpeas are rich in proteins, carbohydrates, minerals, fibers, vitamins, and phytochemicals such as anthocyanins, alkaloids, flavonoids, phytic acids, saponins, steroids, tannins, catechins, terpenoids, and trypsin inhibitors [58,60].

Chickpeas are traditionally used for constipation, diarrhea, dyspepsia, flatulence, sunstroke, weight loss, and many other diseases [61]. The regular consumption of pulses, such as chickpeas, is recommended as part of a healthy lifestyle. Chickpeas are utilized as a dietary supplement in cases of metabolic syndrome and obesity as they contain several bioactive compounds, such as flavonoids and anthocyanins [58]. Chickpeas have antioxidant, antidiarrheal, antidiabetic, anti-inflammatory, anticonvulsant, hepatoprotective, antimicrobial, and many other pharmacological effects [61].

Chickpeas contain a variety of antioxidant phenolic compounds, such as flavonoids and oligomeric and polymeric proanthocyanidins [62]. Chickpeas are rich in proteins such as globulins (57%), glutelins (18%), albumins (12%), and prolamins (3%). Proteins are an important source of peptides that can be released by hydrolysis, with beneficial health effects such as antioxidants (e.g., histidine) and antidiabetic activity by regulating insulin resistance and lipid metabolism [56]. Chickpeas also contain sulfur amino acids (e.g., methionine, cysteine), which have been reported to have antioxidant and anti-inflammatory properties that may aid DM via the attenuation of oxidative stress [60]. Such antioxidant properties of chickpeas are also promising in improving glycemic control and total serum glucose concentrations [56]. The phenolic compounds present in chickpeas exert antidiabetic activity by inhibiting carbohydrate hydrolyzing enzymes such as α-amylase and α-glucosidase, thus contributing to the management of T2DM [62]. Chickpeas have a low ‘glycemic index’, which plays a crucial role in glycemic regulation and insulin secretion. The starch present in chickpeas is very resistant to intestinal digestion. This results in a lower availability of glucose to enter the bloodstream, leading to a reduced demand for insulin [63]. The flavonoids in chickpeas have been reported to increase insulin receptor activity, while phytosterol can stimulate insulin secretion from the pancreas [58]. The isoflavones genistein, biochanin A, and formononetin have been reported to elicit antidiabetic and anti-hyperlipidemic effects in diabetic rats [64].

A clinical study conducted for 8 weeks on 30 obese subjects consuming a legume-restricted vs. legume-based diet revealed that the legume-based diet significantly reduces body weight, LDL cholesterol levels, systolic blood pressure, and total cholesterol levels [63]. Another study found that the long-term oral administration of chickpea extract at a dose of 200 mg/kg body weight has a notable hypoglycemic effect together with the inhibition of free radical production, lipid peroxidation, and the activation of antioxidant enzymes in the liver and kidneys of STZ-induced diabetic rats [58].

### 4.6. Cinnamomum verum J. Presl.

The true cinnamon, also known as Sri Lanka or Ceylon cinnamon, is a brown-colored spice with a delicate aroma and a warm sweet flavor that consists of the dried inner bark of *Cinnamomum verum* J. Presl (formerly called *Cinnamomum zeylanicum*) (Lauraceae). Among the 250 species of Cinnamomum, *C. loureirii* (Vietnamese cinnamon), *C. burmannii* (Indonesian cinnamon), and *C. aromaticum* (Cassia or Chinese cinnamon) are also used as cinnamon [65,66,67,68].

Traditionally cinnamon has been used for cooking and flavoring beverages as well as medicinally to treat arthritis, diarrhea, toothaches, bad breath, digestion, flatulence, piles, amoebiasis, heart diseases, fever, cough, colds, headaches, and menstrual irregularities. It is also known to have antidiabetic properties along with anti-inflammatory, antibacterial, and antioxidant activity. It is commonly used as a dried powder or as an aqueous extract [67,68,69].

The bark of cinnamon contains an essential oil, with 60–80% cinnamaldehyde as the main component. Other minor phytoconstituents are o-methoxycinnamaldehyde, trans-cinnamic acid, vitamins, eugenol, monoterpenoids, procyanidins, diterpenes, phenylpropanoids, mucilage, tannins, flavonoids, glycosides, terpenoids, coumarins, anthraquinones, and polysaccharides. The leaves also contain 70–90% of eugenol [65]. Trace elements such as calcium, chromium, copper, iodine, iron, manganese, phosphorus, potassium, and zinc have also been isolated from cinnamon [66]. Its polyphenolic compounds have been reported to increase glucose entry into cells by improving insulin receptor phosphorylation and the translocation of glucose transporter-4 (GLUT4) to the plasma membrane [69].

Cinnamon has been shown to possess antidiabetic properties via its ability to reduce the digestion and absorption of carbohydrates [68]. The cinnamon extract can inhibit gastrointestinal enzymes, modulate insulin response and sensitivity, improve glucose uptake, and inhibit gluconeogenesis. Cinnamon can reduce the formation of advanced-glycated end products in the gastrointestinal lumen during digestion, therefore, minimizing diabetes complications by reducing oxidative stress, inflammation, and islet cell injury [70].

Cinnamon has been reported to exert a beneficial effect on DM in multiple ways. It has insulin secretory activity [67]. It can also increase the expression of peroxisome proliferator-activated receptors (PPAR) α and γ, as well as increase insulin sensitivity. Furthermore, it can inhibit the intestinal glucosidase and the pancreatic amylase enzymes, thus delaying the gastric emptying time and postprandial glucose concentrations [69].

It contains antioxidants that can inhibit ROS generation and prevent diabetes at the prediabetes state [68]. It also contains double-linked procyanidin type-A polymers, primarily composed of flavonoids such as catechin and epicatechin. These polymers are typically found as trimers and tetramers and possess health-promoting properties, including antidiabetic effects [66,71]. It also contains polyphenol type-A polymers with insulinotropic activity that can inhibit tyrosine phosphatase leading to the inhibition of the de-phosphorylation and activation of the phosphorylation of insulin [67,68]. The hydroxycinnamic acids from *Cinnamomum aromaticum* have been identified as active ingredients with high glucose transport activity. In their esterified form, they increase glucose transport via GLUT4 translocation and enhance the phosphorylation of IR-ß (Insulin Receptor β) and IRS-1 (Insulin Receptor Substrate-1) in adipocytes [71]. One study showed that cinnamaldehyde could lower blood glucose and glycosylated hemoglobin concentrations, as well as significantly increase plasma insulin and hepatic glycogen concentrations and restore altered plasma enzymes (aspartate aminotransferase, alanine aminotransferase, lactate dehydrogenase, alkaline phosphatase, and acid phosphatase) back to normal levels [67]. In another study, a significant reduction in fasting blood glucose and glycosylated hemoglobin levels was observed in T2DM patients after 6 to 12 weeks of daily treatment with 1g of cinnamon. There was also an improvement in the oxidative stress markers [69]. Another investigation showed that the administration of *C. aromaticum* bark extracts at a dose of 200 mg/kg body weight for 6 weeks could notably lower blood glucose, TGs, and total cholesterol concentrations, as well as inhibit intestinal α-glucosidase in diabetic mice [68].

### 4.7. Crocus sativus L.

*Crocus sativus* (Iridaceae) is a perennial herb with significant importance in medicine, cosmetics, food, and hygiene. It is commonly referred to as “saffron crocus” since it is derived from the dried stigmas of the crocus flower. The latter is widely cultivated in countries with mild to dry climates. [72,73]. Saffron petals are also being used for their curative properties [74]. Saffron is reputed for its diaphoretic, expectorant, eupeptic, abortifacient, tranquilizer, aphrodisiac, and emmenagogue properties and is traditionally used for the treatment of hepatic disorders, flatulence, spasms, insomnia, vomiting, dental and gingival pain, depression, seizures, cognitive disorders, lumbago, asthma, cough, bronchitis, colds, fever, cardiovascular disorders, cancer and metabolic syndromes like diabetes and hyperlipidemia [75,76].

Saffron contains a vast array of phytochemicals, including the carotenoids crocetin, crocin (crocetin glycoside), and picrocrocin. Saffron also comprises safranal and over 150 other aroma-yielding volatile compounds [77]. Saffron petals also contain flavonoids (kaempferol), carotenoids, anthocyanins, phenolic compounds, terpenoids, alkaloids, protein, fibers, sodium, potassium, calcium, copper, iron, magnesium, zinc, and phosphorus [74].

Crocin, crocetin, and safranal are the three main active compounds of saffron that have potent antidiabetic effects. The distinct coloration of saffron is primarily attributed to the presence of crocin, while safranal is responsible for its unique aroma [76]. The antidiabetic effects of these components have been demonstrated via their inhibition of free radical chain reactions and their ability to stabilize biological membranes, scavenge reactive oxygen species, and reduce the peroxidation of unsaturated membrane lipids [75]. Moreover, crocin has been reported to significantly reduce plasma levels of TGs and total cholesterol by activating low to moderate PPARα, together with improving high glucose levels, insulin resistance, and atherosclerosis [76]. It can stimulate the islet cells of Langerhans to increase insulin secretion and ameliorate diabetic neuropathy. It also exerts blood glucose-lowering effects by sensitizing the insulin receptors in peripheral muscles [78]. The phosphorylation of mitogen-activated protein kinases (MAPK) and Acetyl-CoA carboxylase (ACC) by saffron bioactive constituents has been found to significantly improve peripheral insulin sensitivity [79].

An ethanol extract of saffron has been reported to decrease TG, TC, and LDL and increase HDL levels [76]. Previous studies showed that oral administration of saffron extract could improve serum levels, body weight, lipid profile, and blood glucose, as well as the augmentation of kidney and liver functions in alloxan-induced diabetic rats [72]. Another study demonstrated that saffron extracts, at doses of 200, 400, and 600 mg/kg, cause significant improvements in serum insulin levels and noticeable reductions in blood glucose levels in diabetic rats [75]. A methanolic extract of saffron, containing crocin and safranal, has been reported to notably reduce fasting blood glucose and HbA1c levels in alloxan-induced diabetic rats [80].

### 4.8. Cuminum cyminum L.

*Cuminum cyminum* (cumin) (Apiaceae) is the most abundant herb cultivated worldwide, after black pepper. It is cultivated in many countries, especially those with semi-arid climates, such as China, India, Iran, Egypt, Saudi Arabia, and the Mediterranean [81,82]. Cumin seeds are utilized for culinary and medicinal purposes. Traditionally, they are commonly employed for treating chronic diarrhea, dyspepsia, asthma, hypertension, fever, inflammation, bronchitis, dizziness, eczema, gastrointestinal disturbances, and diabetes [83]. They are also renowned for their antispasmodic, abortifacient, diuretic, emmenagogue, carminative, and stomachic properties [84].

Cumin possesses highly valued phytochemicals such as terpenes (β-pinene), phenols (eugenol), alcohols and aldehydes (cuminaldehyde), flavonoids (luteolin, catechin, quercetin and apigenin), alkaloids, coumarins, anthraquinones, saponins, tannins, steroids, as well as proteins, resins, fibers, fats (especially monounsaturated), vitamins and minerals. Phenolic acids (gallic, cinnamic, rosmarinic, coumaric, and vanillic acids) are also present in cumin seeds [83,85].

Cumin seeds have been shown to significantly reduce body weight and lower blood glucose, glycosylated hemoglobin, phospholipid, cholesterol, free fatty acid, and TG levels from the plasma and tissues [83]. They are also effective in exerting antioxidant protective effects on insulin-secreting β cells and enhancing insulin secretion. Cuminaldehyde is the main active constituent with potential antidiabetic properties which has inhibitory effects on aldose reductase and α-glucosidase. It also exhibits insulinotropic effects by blocking ATP-sensitive potassium channels and increasing intracellular calcium concentration in pancreatic β cells together with possessing protective effects on pancreatic β cells [86]. Cumin extract has been reported to suppress α-amylase activity, elevated ROS generation, and oxidative injury by optimizing rapamycin (mTOR), surviving, and beclin-1 (BECN1) levels. Cumin also promotes catalase, glutathione reductase, and peroxidase activities and increases ascorbic acid levels, which protect cells against damage caused by oxidative stress [85]. Cuminaldehyde, and the flavonoids present in cumin, not only counteract the damage caused by oxidative stress but can also prevent AGE formation, which is involved in the pathogenesis of diabetic microvascular complications [81].

One clinical study reported that green cumin (50 or 100 mg/kg/day) could notably improve hyperglycemia and insulin sensitivity after 8 weeks of treatment in T2DM subjects [86]. A similar study with cumin seeds has reported that cumin could improve fasting blood glucose and glycosylated hemoglobin levels and inhibit α-amylase activity [82].

### 4.9. Eugenia caryophyllata Thunb.

*Eugenia caryophyllata* (Myrtaceae) is a tree native to Indonesia and is now cultivated in many countries worldwide. *Syzygium aromaticum* and *Eugenia caryophyllata* are synonyms. The spice known as clove corresponds to the dried unopened flower buds of the plant. Clove is used for culinary purposes and ethnomedicine [87,88,89]. The stems, leaves, and fruits of *E. caryophyllata* can also be used for medicinal and cooking purposes [87]. Clove is reputed for its beneficial effects on a variety of ailments, including dental problems, nausea, liver, bowel and stomach disorders, vomiting, flatulence, scabies, cholera, malaria, tuberculosis, bacterial and protozoal infections, and food-borne pathogens. Clove can also act as a stimulant for the nerves [88,90].

The main phytoconstituents isolated from the clove are phenolic volatile compounds such as eugenol (78%), β-caryophyllene (13%), α-humulene, caryophyllene oxide, and acetyleugenol [91,92]. Clove also contains tetraethylammonium chloride, gallic acid, phenolic acids, polyphenols, and flavonol glycosides with free radical scavenging/antioxidant activity [89]. Isoeugenol has proven to be beneficial for T2DM management, owing to its inhibitory effects against α-glycosidase and α-amylase [93].

A clove extract has been reported to act like insulin by reducing phosphoenolpyruvate carboxykinase (PEPCK) and glucose-6-phosphatase (G6Pase) gene expression in hepatocytes and hepatoma cells [87]. Tannins, flavonoids, ellagic acid, gallic acid, and their glycosides isolated from an alcoholic and aqueous extract of clove buds have also been reported to exert hypoglycemic activity [90]. Studies have also revealed that the dietary supplementation of cloves significantly reduces elevated blood sugar levels and lipid peroxidation in STZ-induced diabetic rats while restoring the antioxidant enzymatic level [91]. Another in vivo experiment demonstrated that eugenol, at a dose of 100 mg/kg for 4 days, suppresses the oxidative stress caused by gentamicin and was effective in dyslipidemia [90].

### 4.10. Foeniculum vulgare Mill.

*Foeniculum vulgare* (Umbelliferae or Apiaceae) is an aromatic Mediterranean plant commonly known as fennel. Its seeds are used as a flavoring agent in food or traditional medicine and consumed as a medicinal tea. Traditionally, fennel has been used as an appetite suppressant to ease childbirth, increase milk secretion, promote menstruation, ease male menopause, and increase libido [94,95,96,97].

Fennel seeds contain an essential oil rich in trans-anethole, fenchone, limonene, camphor, and α-pinene. They also contain a fixed oil with free fatty acids (petroselinic acid and oleic acid) and tocopherols [95].

Fennel exerts antitumor, antioxidant, cytoprotective, cancer chemopreventive, hepatoprotective, and hypoglycemic activities [98]. Fennel oil possesses pro-oxidant, antioxidant, and anti-inflammatory effects [95]. Fennel seeds can inhibit glutathione peroxidase (GSHPx), α-amylase, and α-glucosidase, thus delaying the breakdown of carbohydrates [96].

An extract prepared from fennel mixed with Cassia angustifolia has been reported to improve body weight, serum cholesterol, TGs, LDL, HDL, oxidative stress, MDA, SOD, CAT, and GSH levels in STZ-induced diabetic rats. The hypoglycemic effect of fennel seeds may be mediated via the preservation of pancreatic β-cell integrity [98]. The essential oil of *F. vulgare* ameliorated hyperglycemia, glutathione levels, and kidney and pancreas functions in STZ-induced diabetic rats [95]. An extract from fennel seeds, and its main compound trans-anethole, were reported to improve liver tissue damage, liver enzyme function, blood glucose levels, lipid profile, body weight, and food and fluid intake in STZ-induced diabetic rats [96].

### 4.11. Hordeum vulgare L.

*Hordeum vulgare* (Poaceae), also known as barley, is one of the highest dietary fiber-containing crops and is currently the fourth most cultivated cereal crop in the world [99]. It is also one of the most cultivated grains, particularly in Eurasia. Barley is a popular and affordable dietary source of high fiber. Its grains, leaves, and sprouts are used for their antioxidant, hypolipidemic, antidepressant, and antidiabetic effects. Barley is also used for skin abnormalities, arthritis, digestive diseases, weight loss, cancer, and its detoxifying properties [100,101,102].

Barley is mainly rich in β-glucan, starch, proteins, minerals, and soluble fiber [103]. It also contains arabinoxylans, phenolics (derivatives of ferulic acid, vanillic acid, syringic acid, and p-coumaric acid), flavonoids (flavonols and anthocyanins), tocols (tocopherols and tocotrienols), lignans, phytosterols and folates [104].

Its soluble fibers possess beneficial effects on metabolic syndromes, lipid metabolism, high blood glucose levels, and bowel function [101]. Among its minerals, K+ plays an important role in preventing Alzheimer’s disease and hypertension and lowering oxidative stress. Furthermore, the presence of sulfide and quercetin also helps to reduce inflammation and obesity and to aid heart diseases and diabetes [99]. β-glucan, the main bioactive chemical in barley, has been demonstrated to lower total cholesterol by inhibiting hepatic cholesterol synthesis. Soluble fiber from barley undergoes fermentation in the colon and further leads to the formation of small-chain fatty acids, which are absorbed easily and are able to inhibit hepatic cholesterol synthesis [103]. Barley is a potent α-glucosidase inhibitor. It can also reduce the postprandial glucose response and improve insulin resistance. Its vitamins, minerals, β-glucan, phenolics, and flavonoids (ferulic acid, naringin, and catechin) have demonstrated hypoglycemic activity by inhibiting α-glucosidase and α-amylase [104].

One study has reported that barley extract significantly reduces blood glucose levels in diabetic rats but not normal rats. The mechanism is similar to that of insulin secretagogues such as biguanides or α-glucosidase inhibitors, resulting in decreased insulin resistance and interfering with carbohydrates absorption or metabolism [101]. Another study revealed that the repeated consumption of 10% *w*/*v* barley water and its components, such as amino acids, in alloxan-induced diabetic rats could restore changes in the immunological and biochemical parameters to their normal levels, indicating antidiabetic activity [102].

### 4.12. Juglans regia L.

*Juglans regia* (Juglandaceae), commonly known as walnut, is a plant with numerous therapeutic benefits in traditional medicine [105]. Traditionally, walnut fruits and leaves have been employed in pharmaceuticals as astringent, antiseptic, and anti-hyperglycemic agents to mitigate diabetic complications [106,107].

Walnut leaf, fruit, and flower are rich in phytoconstituents such as vitamin C, vitamin E, β-carotene, lipoic acid, quercetin, naphthoquinones, flavonoids (quercetin), gallic acid, polyphenols (caffeoylquinic acid), linoleic and linolenic acids, tannins, and folates [108,109].

Several of these constituents have strong antioxidant/free-radical scavenging activity, along with antidiabetic effects. Walnut is able to decrease blood glucose levels with the phenols present in their leaves. It has also been reported to increase insulin and reduce HbA1c levels in T2DM patients via mechanisms similar to those of metformin and glibenclamide. Walnut leaf extract exerts antioxidant, anti-inflammatory, and anti-apoptotic effects, which help with diabetes complications. The flavonoids (e.g., quercetin) can actively decrease blood glucose levels by inhibiting the GLUT2 glucose transporter, suppressing glucose intestinal absorption. Phenolic acids (e.g., caffeoylquinic acid) can inhibit glucose-6-phosphate translocase, leading to a reduction in hepatic glucose production and lowering blood glucose and HbA1C levels [106,108,109].

One study revealed that walnut bark extracts alleviate DM complications, while leaves and fleshy green fruits improve blood glucose. An infusion of walnut and olive leaves has been reported as a good combination of plants to lower blood glucose [105]. Clinical studies have confirmed that walnut is a strong hypoglycemic agent, lowering fasting blood glucose, HbA1c, cholesterol, and TGs while increasing insulin levels with few side effects [106].

### 4.13. Lens culinaris L.

*Lens culinaris* (Leguminosae), also known as lentils, is a widely cultivated pulse crop worldwide with notable health benefits [110,111]. Lentil seeds and sprouts have a high nutritional and medicinal value. They are rich in quality fiber, protein, and starch and are low in fat [112]. Due to their high protein content, lentils have been used traditionally as a meat substitute among poor people [113].

Lentils contain polyphenols, phytates, triterpenoids, defensins, phytosterols, flavonoids, saponins, protease inhibitors, dietary fibers, and lectins, among which saponins and polyphenols are potent radical scavengers and HMG-CoA reductase inhibitors [114].

The regular consumption of lentils has been reported to lower the glycemic load, fasting blood sugar, and glycemic index while improving the lipid profile and lipoprotein metabolism in both diabetic and healthy individuals [113]. Interestingly, the enzymatic hydrolysis of lentils increases their antioxidant/free-radical scavenging and antidiabetic effects [115]. Several phytoconstituents have been reported to exert preventive and therapeutic effects on chronic diabetes and hyperlipidemia. Polyphenols and flavonoids have been shown to contribute to the antidiabetic, anti-obesity, antioxidant, and anti-inflammatory properties of lentils [113,114]. The presence of fiber has also been shown to help improve metabolic impairment and glycemic control in both T1DM and T2DM patients. Along with a glucose-lowering effect, the lentils can reduce glucose poisoning or the damaging effects of glucose associated with the destruction of pancreatic β cells [116]. The polyphenols, flavonoids, and fiber content in lentil seeds altogether play a significant role in promoting gut motility and preventing metabolic impairment in diabetic rats [113]. Phenolic compounds have an effect on α-glucosidase and lipid digestion, thus helping to maintain glucose and lipid homeostasis [117].

### 4.14. Nigella sativa L.

*Nigella sativa* L. (Ranunculaceae), also known as black cumin, is a medicinal plant used for a variety of ailments, including digestive disorders, dyspepsia, dyspnea, chronic diarrhea, blotting, colds, spider bites, toothaches, warts, headaches, and diabetes. Its seeds are commonly used as a spice [81,118,119,120,121].

Black cumin seeds contain a volatile oil rich in low-molecular-weight terpenoids (carvone, thymoquinone, and thymol) and a fixed oil rich in fatty acids (linoleic acid, oleic acid, palmitic acid). They also contain mucilage, alkaloids, organic acids, tannins, reducing sugars, saponins, resins, phytosterols and steryl esters, amino acids, as well as vitamins and minerals [120,122]. Black cumin roots contain antidiabetic phenolic compounds [121]. The thymoquinone present in *N. sativa* can lower blood glucose levels via extra pancreatic actions [122].

Black cumin seeds have been reported to possess antimicrobial, antipyretic, spasmolytic, antioxidant, antihypertensive, anti-inflammatory, antihistaminic, antitumor, antifertility, antibacterial, cardiovascular, and hypoglycemic properties [118,121]. *N. sativa* also has insulinotropic properties via the maintenance of pancreas β-cell integrity. The oil from its seeds (which contains high amounts of thymoquinone) can significantly decrease serum glucose, LDL cholesterol, TGs, total cholesterol, alanine aminotransferase, and aspartate aminotransferase levels [123]. Black cumin seeds have been reported to reduce the plasma concentrations of glucose, cholesterol, and TGs [121]. Black cumin seeds also induce the proliferation of pancreatic β-cells, increase insulin secretion, activate the AMPK pathway, and stimulate glucose uptake in skeletal muscle cells and adipocytes [119]. An intake of black cumin seeds (2 g twice daily) has been reported to decrease the blood glucose levels of diabetic individuals [122].

Black cumin seed oil administered to STZ- and nicotinamide-induced diabetic hamsters improved blood glucose and serum albumin levels via insulinotropic effects on pancreatic β-cells. The volatile oil administered intraperitoneally produced a notable hypoglycemic effect in alloxan-induced diabetic rabbits [122]. In another study, an ethanol extract of *N. sativa* seeds was found to enhance insulin secretion from pancreatic β-cells and increase glucose uptake in muscle and fat cells. After 18 h of treatment, the extract enhanced glucose-stimulated insulin secretion by more than 35% without affecting glucose sensitivity and accelerated β-cell proliferation. Basal glucose uptake was increased by 55% in muscle cells and approximately 400% in adipocytes, indicating that black cumin seeds have insulinotropic and insulin-like properties [119]. The active constituent of *N. sativa* seeds, thymoquinone showed potent radical scavenging activity both in vitro and in vivo, which can be beneficial in reducing T2DM-associated oxidative stress [122].

### 4.15. Olea europaea L.

The olive, *Olea europaea* L. (Oleaceae), is best known for its fruits, which yield olive oil when mechanically pressed [124,125]. The plant is extensively cultivated and used for cooking, cosmetics, and medicinal purposes. It is native to many countries in the world except for tropical, subtropical, and hot temperate regions. It is commonly found in the Mediterranean zone [124].

The fruit pit contains 20–30% oil. The latter is rich in flavonoids, iridoids, triterpenes (oleanolic acid), benzoic acid derivatives, secoiridoids (oleuropein), phenolic compounds, isochromans, and other classes of secondary metabolites. Recent studies have reported that oleanolic acid can reduce blood glucose levels and improve glucose tolerance and plasma insulin levels by acting as an agonist on TGR5, bile acid surface receptor [124,125].

Olive oil is traditionally used for hypertension, inflammation, diarrhea, respiratory disorders, urinary tract infections, hemorrhoids, rheumatism, stomach and intestinal disorders, asthma, and diabetes and as a laxative, mouth cleanser, and vasodilator. It is also known to reduce blood sugar, cholesterol, and uric acid levels [124]. The leaves also have notable pharmacological properties such as antioxidant, anti-inflammatory, antiarrythmatic, hypotensive, and immunostimulatory effects [125,126].

Olive oil possesses anti-inflammatory, antioxidant, cardioprotective, anticonvulsant, immunomodulatory, antinociceptive, gastroprotective, analgesic, antimicrobial, antiviral, antihypertensive, anticancer, wound healing, and antihyperglycemic activities [124,127]. The phenol-enriched olive oil has been reported to inhibit α-glucosidase, indicating its potential in the management of T2DM [128]. Hydroxytyrosol and oleuropein from olive oil exert antihyperglycemic activity via the inhibition of α-glucosidase and act as antioxidant agents [129]. Oleanolic acid, one of the main active phytoconstituents of olive oil, has also been demonstrated to improve insulin action and sensitivity and promote β-cells survival and proliferation via the inhibition of cytokine production in STZ-induced diabetic mice [130]. A polyphenol-rich extract of olive oil showed antioxidant properties and preserved cellular GSH levels, which may alleviate oxidative damage in T2DM patients [131].

### 4.16. Pinus gerardiana Wall. ex D. Don

*Pinus gerardiana* (Pinaceae) is also known as chilgoza pine, pine nut, or chilgoza seed [132]. Pine nuts are used as food, either raw or roasted, and in traditional medicine as a diuretic, expectorant, antibacterial, antiseptic, antifungal, antihypertensive, antiviral, and antineuralgic agent [133].

Nuts are rich in fatty acids (stearic acid, gallic acid, ellagic acid, linolenic acid, oleic acid, arachidic acid, and palmitic acid), tocopherols, carotenoids, phytosterols, carbohydrates, proteins, minerals, and vitamins [132,134].

Chilgoza nuts have been reported to promote weight loss, reduce oxidative stress, fasting blood glucose levels, and malondialdehyde, and increase total thiol groups, superoxide dismutase enzyme activity, and the total antioxidant capacity of serums and livers [135]. Gallic acid and ellagic acid, two phenolic compounds present in chilgoza, can decrease the expression of the PPARγ gene and activate Akt (protein kinase B). They also protect pancreatic β-cells, induce insulin secretion, and reduce glucose intolerance, thus helping in the management of DM [135]. A pine nut extract has been demonstrated to lower blood glucose levels, reduce body weight and hyperlipidemia, and improve liver and kidney functions in STZ-induced diabetic rats. Its beneficial effect in DM has been attributed to the presence of phenolic compounds and flavonoids. These compounds also contribute a normoglycemic effect via α-amylase inhibition and act as antioxidant agents, thus preventing diabetic complications. They also play a role in regulating glycolytic and gluconeogenic activities, including hexokinase, glucose-6-phosphatase, fructose-1,6-bisphosphatase, and glycogen phosphorylase [136].

### 4.17. Piper nigrum L.

*Piper nigrum* (Piperaceae), also known as black pepper, is a plant native to India and other tropical countries. Its single round-shaped seed is commonly used for culinary, preservation, flavoring, and medicinal purposes. *P. nigrum* (seeds, flowers, fruits, and leaves) is also widely used for its ethnomedicinal properties, particularly in Asia. The plant is reputed to relieve pain, menstrual problems, atrophic arthritis, digestive problems, apathy, influenza, and fever as a nerve tonic [137,138,139,140].

Black pepper contains an essential oil that is rich in piperine. Other isolated compounds found in the oil are β-caryophyllene, limonene, sabinene, α-pinene, β-bisabolene, α-copaene, α-cadinol, α-thujene, and α-humulene. Black pepper leaves are rich in nerolidol and α-pinene [141]. Other secondary metabolites found in black pepper are alkaloids (e.g., piperine), glycosides, terpenoids, steroids, flavonoids, tannins, and anthraquinones [137].

Black pepper possesses antibacterial, anti-inflammatory, antipyretic, anti-snake venom, antiplatelet, antihypertensive, anticancer, antioxidant, analgesic, antidepressant, antidiarrheal, and antidiabetic properties [141]. Phytoconstituents such as piperine isolated from black pepper also exhibit antioxidant properties. Moreover, a combination of piperine and hydroxytyrosol has been reported to exhibit radical scavenging properties, increase insulin secretion, improve white adipose tissue formation, lower blood glucose, lipid peroxidation, and lipogenesis via the modulation of the transcription factors NF-κB, Nrf2, SREBP-1c, and PPAR-γ as well as their target genes [141,142,143]. Piperine has been shown to reduce lipid peroxidation while activating antioxidant enzymes in diabetic animals [144]. One study revealed that a combination of black pepper, turmeric, and palm dates increases insulin and HDL levels, lowers blood glucose, TGs, total cholesterol, and LDL levels, and shows antioxidant capacity in diabetic rats, indicating the potential of black pepper in the management of DM [143].

### 4.18. Pistacia vera L.

*Pistacia vera* (Anacardiaceae) has significant economic importance worldwide. The plant is commonly known as pistachios, used for its fruits, seeds, and leaves for both culinary and medicinal purposes [145,146]. The shells, roots, and stems parts also contain pharmacologically important constituents [147]. Pistachios are native to Asia and the Mediterranean region [148]. Traditionally, pistachios have been used for their diuretic, antidiabetic, and anti-inflammatory properties, as well as for strengthening gums, coughs, chills, asthma, stomach aches, abscesses, bruises, itching and sores, chest ailments, rheumatism, gynecological issues, and as a topical remedy for hemorrhoids [146,149].

Pistachios contain oleoresins, polyphenols, triterpenoids, polymeric procyanidins, flavonols, anthocyanins, and pistacionic acids [150].

Studies have indicated that the Pistacia genus possesses various biological activities, including anti-inflammatory, antioxidant, antimicrobial, antiviral, anti-osteoarthritis, anti-gout, anti-epileptic, sedative-hypnotic, muscle-relaxant, anti-fatigue, wound-healing (diabetic wound) and second-degree burn healing, anti-colitis, anti-peptic ulcer, neuroprotective, hypoglycemic, hepatoprotective, lipid-lowering, anti-obesity, nephroprotective, and antidiabetic properties [145]. Pistacia has also demonstrated strong antihypertensive effects via inhibiting the angiotensin-converting enzyme-1, thereby providing beneficial effects in the cardiovascular complications associated with DM [145].

One study revealed that pistacionic acid exhibits antidiabetic activity by inhibiting α-glucosidase and α-amylase [147]. Another study demonstrated that *P. vera* has inhibitory activity on α-amylase, α-glucosidase, pancreatic lipase, and cholesterol esterase enzymes involved in carbohydrate digestion, thereby delaying carbohydrate digestion, lowering glucose absorption, and decreasing blood glucose levels. Moreover, *P. vera* extract is also reported to exhibit antioxidant activity in in vitro models, which may reduce oxidative stress and contribute to the improvement of T2DM [149].

### 4.19. Vitis vinifera L.

*Vitis vinifera* (Vitaceae) is a plant identified as a wine grape or raisin. Traditionally, grapes have been valued for their use in winemaking. They also have nutritional benefits and the potential as a functional food. The dried, seedless grapes produce raisins, which are consumed widely for their health-promoting effects [151,152,153].

Raisins are rich in polyphenols (e.g., kaempferol, quercetin, and ellagic acid), phenolic acids, caftaric acid, citric acid, vanillic acid, stilbenes (e.g., trans-resveratrol), protocatechuic, ferulic, caffeic, gallic, syringic, and p-coumaric acids. Raisins are also rich in dietary fiber [154].

The pharmacological properties of raisins include antioxidant, antidiabetic, immunomodulatory, neuroprotective, anticarcinogenic, anti-obesity, and anti-aging effects [155]. The consumption of raisins has been linked to improvements in blood glucose control, reduction in postprandial glucose, HbA1c, and blood pressure levels. Furthermore, raisins are rich in antioxidant compounds (e.g., ellagic acid, trans-resveratrol), which can inhibit LDL cholesterol oxidation by scavenging free radicals and improve metabolic syndromes such as DM [152,155].

One study found that a 6-month dietary intervention using raisins resulted in a significant increase in the plasma total antioxidant potential of T2DM patients. Additionally, fiber-rich raisin also exhibits potent hypocholesterolemic activity [154].

### 4.20. Zingiber officinale Roscoe

*Zingiber officinale* (Zingiberaceae), also known as ginger, is a medicinal plant native to Asia. Its roots and rhizomes are used as a spice and flavoring agent and for medicinal purposes [156,157,158]. Traditionally, ginger has been used to treat digestive disorders, nausea, rheumatism, respiratory conditions, cough, bleeding, baldness, toothache, diabetes, hypercholesterolemia, neurological diseases, asthma, stroke, constipation, and cancer [158,159].

Ginger mainly contains monoterpenes, including oxygenated monoterpenes, such as geranial, linalool, borneol, citronellal, neral, and α-terpineol, as well as sesquiterpenes including zingiberene, zingiberenol, α- and β-farnesene, ar-curcumene, copaene, and cadinene [156].

The gingerols, shogaols, and volatile oils are the main contributors to the pharmacological effects of ginger [159]. Its antidiabetic properties are attributed to phenolic (diarylheptanoids and diarylheptanoid-derived gingerols and shogaols), as well as non-phenolic compounds (sesquiterpene and monoterpene hydrocarbons, carbonyl compounds, and esters). Ginger also exhibits strong antioxidant/radical scavenging activity, therefore, mitigating diabetic complications [158,160,161]. Ginger has been reported to strongly inhibit α-glucosidase while mildly inhibiting α-amylase as well as effectively reduce serum glucose, cholesterol, and TGs levels together with eliminating proteinuria associated with diabetic nephropathy [159,162]. It can also increase HDL cholesterol concentrations and improve insulin sensitivity [44,157].

One study showed that a hydroalcoholic extract of ginger at a dose of 400 mg/kg lowers blood glucose and increases body weight and serum insulin in STZ-induced diabetic rats [156]. Gingerols were found to inhibit both prostaglandin and leukotriene biosynthesis as well as angiogenesis, which may attenuate inflammation associated with diabetes [159]. Another study in STZ-induced diabetic rats showed that ginger significantly lowers blood glucose levels and decreases oxidative stress and body weight, thus demonstrating antidiabetic and antihyperlipidaemic roles [160].

The summary of the pharmacology of 20 common medicinal plant-based diets, specific antidiabetic phytoconstituents, and their chemical structures have been shown in Table 1, Table 2 and Table 3, respectively.

## 5. Discussion

DM is a disease primarily attributed to a deficiency in insulin production or action, with oxidative stress and inflammation being the main mediators of its progression [16,17,185]. Several epidemiological studies have shown that dietary habits have a significant impact on the prevention of diseases, with plant-derived constituents in vegetables, fruits, spices, and condiments possessing beneficial health properties (e.g., antioxidant, immunomodulatory, anti-hyperlipidemic, anti-inflammatory, and anti-hyperglycemic effects) [27,28]. In the typical Western diet, which comprises mostly processed foods, red meat, and fast-acting carbohydrates, these phytoconstituents are lacking, and this has been demonstrated to contribute to the development of DM [1,4]. Understanding the relevance of dietary plant-based constituents to DM, including their pharmacological properties and mode of action, can be an effective strategy to better manage and prevent DM, potentially reducing the demand for medications and preventing diabetic complications [6,11].

Vegetables (e.g., cabbage, lentils, onions), fruits (e.g., grapes), herbs (e.g., dill, thyme), spices (e.g., black pepper, cinnamon, garlic, ginger, cumin), and nuts (e.g., walnuts, pistachios, pine nuts) contain a wide variety of phytochemicals (e.g., flavonoids, anthocyanins, saponins, tannins and carotenoids) that have been shown to possess antidiabetic properties. Olive oil and honey also provide natural chemicals that have demonstrated antidiabetic activity. The aforementioned plant-derived foods exert their antidiabetic activity on multiple organs (e.g., liver, intestine, pancreas, skeletal muscle, adipose tissue) and via different mechanistic pathways [28,141,156,168] (Figure 1).

Previous studies have established the antidiabetic and antihyperlipidemic effects of a diverse range of phytochemicals in plant-based foods. This includes the organosulfur compound allicin from garlic, flavonoids (genistein, formononetin, biochanin A and quercetin) from chickpeas, cinnamaldehyde from cinnamon, isothiocyanates and anthocyanidins from cabbage, carotenoids (crocetin, crocin) and safranal from saffron, thymoquinone from black cumin, organic acid (linolenic acid, oleic acid, arachidic acid and palmitic acid) in chilgoza nuts, procyanidins from pistachios, the sesquiterpene β-caryophyllene and the alkaloid piperine from black pepper, the stilbene resveratrol from grapes and curcuminoids, 6-gingerol, and 6-shogaol from ginger.

In animal models of DM, the consumption of these plant-based food products has been shown to reduce oxidative stress-induced damage and increase insulin secretion as well as exhibit hypolipidemic, hypoglycemic, and anti-inflammatory activities. The consumption of such products also helps to maintain a normal lipid profile, regulate blood glucose levels, inhibit ROS generation, decrease LDL, and increase HDL cholesterol level [74,80,124].

Long-term consumption of plant-based diets may be deficient in essential nutrients such as protein, EPA, DHA, PUFA vitamin B12, vitamin D, iron, zinc, iodine, calcium, and bone turnover markers compared to non-vegetarian diets, which could cause potential deficiencies and health consequences. Furthermore, excessive fiber consumption may lead to mild digestive issues such as bloating, gas, and occasional bouts of diarrhea [186,187]. However, most plant-based diets, such as *Nigella sativa*, have been found to be well tolerated and non-toxic, even at high doses [121]. Therefore, to compensate for these minor challenges maintaining a balance is important.

The vast array of bioactive compounds found in medicinal plants continues to be an important resource for drug discovery and development, and structure–activity relationship (SAR) studies are important to understand how minor modifications in chemical structures can modulate antidiabetic or antioxidant activity [188]. SAR analysis has demonstrated that alkaloids found in medicinal plants such as *Coptis chinensis, Commelina communis*, *Zingiber officinale*, *Nigella sativa, Cuminum cyminum*, *Anemarrhena asphodeloides*, and *Piper nigrum* can improve postprandial hyperglycemia by inhibiting maltase-glucoamylase, which is becoming increasingly important as a target in antidiabetic drug discovery [84,121,137,156,189]. Such studies help researchers to design and develop compounds with improved activity and selectivity and also highlight the role of plant-based foods in the development of new drugs to control diabetes. However, further extensive studies are warranted to better understand the mechanistic pathways through which plants may elicit their antidiabetic effects.

## 6. Conclusions

Many of the medicinal plant-based foods protecting against diabetes mentioned in this review have been safely consumed since ancient times in various parts of the world. There is, however, a lack of sufficient scientific research on their protective effects in humans, particularly understanding the impact of high-dose and long-term consumption on health, and what happens when they are combined with conventional antidiabetic medications as this may lead to unwanted side effects and interactions. Thus, it is important to practice caution when consuming such plants in the context of managing DM. Future studies should focus on conducting high-quality clinical trials to validate the efficacy and establish the therapeutic index of the phytochemicals that have already demonstrated promising antidiabetic activity in vitro and in vivo. Such studies will provide a better understanding of the effectiveness, mechanisms of action, pharmacokinetic, and potential adverse effects of active constituents from medicinal plant-based foods. This has the potential to lead to the development of novel, safer, and more cost-effective plant-based medicines to tackle the rising prevalence of diabetes, particularly in low- and middle-income countries.

## Figures and Tables

**Figure 1 nutrients-15-03266-f001:**
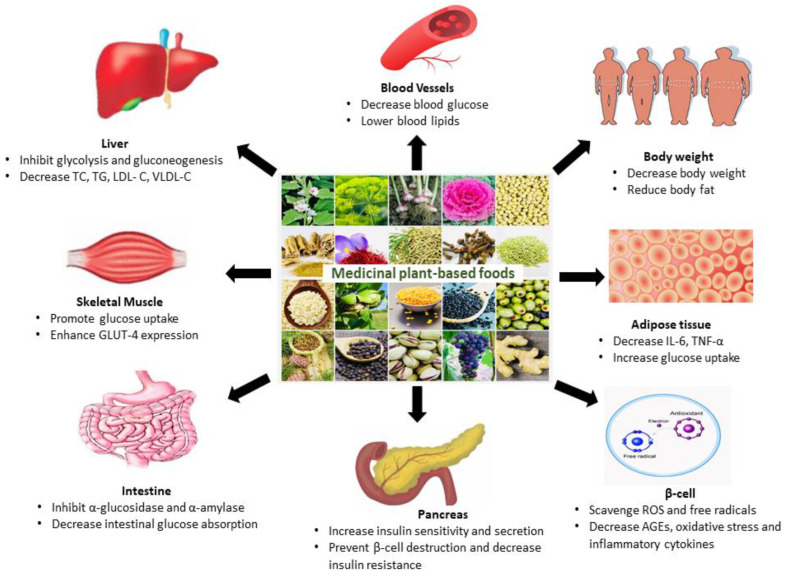
Antidiabetic effects of 20 medicinal plant-based foods on body weight and cells and organs (pancreas, blood vessels, intestine, liver, skeletal muscle, adipose tissue, and β-cells) associated with diabetes. Medicinal plants decrease body weight and body fat by initiating lipolysis; decrease glucose production by inhibiting gluconeogenesis and glycolysis in liver; decrease blood glucose levels by binding to insulin receptor substrate (IRS-1); decrease blood lipid levels by inhibiting HMG-CoA reductase; promote glucose uptake and enhance GLUT-4 expression by activating the AMPK pathway in skeletal muscles; inhibit α-glucosidase and α-amylase enzymes and decrease glucose absorption in the small intestine; improve insulin sensitivity/secretion, improve β-cell function, and lower insulin resistance by activating PPAR-γ expression in the pancreas; decrease IL-6/TNF-α and enhance glucose uptake by activating AMPK in adipose tissues; decrease ROS/free radicals/AGEs, oxidative stress, and inflammatory cytokines in β-cells via antioxidant/radical scavenging activity.

**Table 1 nutrients-15-03266-t001:** Traditional uses and pharmacological effects of the selected antidiabetic medicinal plants.

Medicinal Plants	Plants Parts	Traditional Uses	Recipients	Pharmacological Effects	Dose Administered	Duration of Treatment	References
*Althaea officinalis* L.	Leaves, flowers, roots	Diabetes, inflammation, skin infection, digestive and respiratory disorders	Alloxan-induced diabetic rats	↓ Glutamate pyruvate transaminase (GPT), ↓ cholesterol, ↓ serum glucose levels, ↓ alkaline phosphate level (APL)	Powdered leaves (5% of the diet)	28 days	[33]
*Anethum graveolens* L.	Leaves, stems, and seeds	Diabetes, digestive disorders, cancer, microbial infections, inflammation, hyperlipidemia,	STZ-induced diabetic rats	↓ Inflammatory cytokines, ↓ triglycerides, ↓ total cholesterol, ↓ LDL-C, ↓ VLDL-C, ↓ blood glucose, ↓ AGEs, ↓ protein glycation, ↓ fructosamine level, ↓ fasting blood glucose	300 mg/mL	56 days	[38,39]
*Allium sativum* L.	Pulp	Diabetes, respiratory tract disorders, bacterial and fungal infections, wounds, cancers, CVDs, abdominal discomfort, diarrhea, cold, asthma, hay fever, inflammation, obesity	STZ-induced diabetic rats	↓ LDL, ↓ total cholesterol, ↓ oxidative stress, ↑ HDL, ↓ blood glucose, ↓ intestinal glucose absorption, ↓ ROS generation, ↑ intracellular GSH content, ↓ endothelial dysfunction	200 mg/kg	21 days	[43,44,45]
*Brassica oleracea* L.	Leaves	Inflammation, digestive disorders, cancer, peptic ulcer, gout, detoxification	STZ-induced diabetic rats	↑ Glucose homeostatic regulation, ↓ organ damage from T2DM, ↓ oxidative stress, ↓ obesity, ↓ LDL cholesterol, ↓ total cholesterol, ↓ lipid peroxidation, ↑ pancreatic β-cells functions	250 mg/kg	40 days	[52,53,55]
*Cicer arietinum* L.	Leaves, fruits, seeds	Diabetes, constipation, diarrhea, dyspepsia, flatulence, weight loss, inflammation, microbial infections	STZ-induced diabetic rats	↑ Glycemic control, ↓ total cholesterol, ↓ α-amylase and α-glucosidase activity, ↑ insulin secretion and receptor activity, ↓ insulin deficiency, ↓ LDL	200 mg/kg	30 days	[58,61,62]
*Cinnamomum verum* J. Presl.	Dried bark (inner part)	Diabetes, arthritis, diarrhea, hemorrhoids, toothache, cough, cold, menstrual irregularities, inflammation, bacterial infections	T2DM patients	↓ GI enzymes, ↑ insulin response and sensitivity, ↑ glucose uptake, ↑ glycogen synthesis, ↓ gluconeogenesis, ↓ AGE formation, ↑ phosphorylation, ↑ GLUT4	1000 mg/day	84 days	[67,68,69]
*Crocus sativus* L.	Flower stigma	Diabetes, hepatic and cognitive disorders, lumbago, asthma, cough, bronchitis, CVDs, cancer, hyperlipidemia	Alloxan-induced diabetic rats	↓ ROS, ↓ TG, ↓ TC, ↓ blood glucose, ↓ insulin resistance, ↑ insulin secretion and sensitivity, ↓ LDL, ↑ HDL, ↑ serum insulin, ↓body weight, ↓ lipid level	25, 50, 100, 200 mg/kg	60 days	[75,76]
*Cuminum cyminum* L.	Seeds	Diabetes, chronic diarrhea, dyspepsia, asthma, hypertension, inflammation, bronchitis, dizziness, eczema, gastrointestinal disturbances	T2DM patients	↓ Blood glucose, ↓ glycosylated hemoglobin, ↓ body weight, ↓ phospholipid, ↓ cholesterol, ↓ free fatty acid, ↓ TG, ↑ insulin secretion, ↓ aldose reductase, ↓ α-amylase and ɑ-glucosidase activity, ↓ AGEs	50,100 mg/kg	56 days	[83,84,85,86]
*Eugenia caryophyllata* Thunb.	Unopened dried flower buds, stems, leaves, and fruits	Nausea, hepatic, bowel and stomach disorders, vomiting, microbial and protozoal infections, cholera, malaria, tuberculosis	STZ-induced diabetic rats	↓ PEPCK, ↓ G6Pase gene expression, ↓ AChE, ↓ α-glycosidase, ↓ α-amylase, ↓ elevated blood sugar, ↓ lipid peroxidation	100 mg/kg	105 days	[88,90,91,92]
*Foeniculum vulgare* Mill.	Seeds and fruits	Diabetes, lactation, menstruation irregularities, libido, tumor, inflammatory disease, cancer, hepatic disorders	STZ-induced diabetic male rats	↑ GSH, ↓ α- amylase and α- glucosidase activity, ↓ breakdown of carbohydrates, ↑ glycemic control, ↓ cholesterol, ↓ TG, ↓ LDL, ↑ HDL	150 mg/kg	28 days	[95,96,98]
*Hordeum vulgare* L.	Grains, leaves, sprouts	Diabetes, skin infections, arthritis, digestive diseases, weight loss, cancer, detoxification, lipid metabolism	STZ-induced diabetic rats	↓ Blood glucose, ↓ cholesterol, ↓ hepatic cholesterol synthesis, ↓ α-glucosidase, ↓ α-amylase, ↑ insulin secretion	100, 250, 500 mg/kg	11 days	[101,102,103,104]
*Juglans regia* L.	Husks, kernels, shells, seeds, flowers, barks, and leaves	Diabetes, asthma, arthritis, eczema, stomachache, sinusitis, diarrhea, astringent, antiseptic	T2DM patients	↓ Blood glucose, ↑ insulin, ↓ HbA1c, ↑ GLUT2, ↓ glucose intestinal absorption, ↓ FBG, ↓ TG	100 mg/kg	90 days	[105,106,107]
*Lens culinaris* L.	Seeds and sprouts	Diabetes, meat substitutes, obesity, inflammation, hyperlipidemia	STZ-induced diabetic mice	↓ ROS, ↑ lipoprotein metabolism improvement, ↑ glycemic control, ↓ fasting blood glucose, ↓ serum blood glucose, ↑ gut motility, ↓ body weight	100, 200, 400 mg/kg	21 days	[113,114,115,117]
*Nigella sativa* L.	Seeds	Diabetes, digestive disorders, diarrhea, warts, toothaches, swellings, dyspnea, microbial infections, fever, inflammation, hypertension, allergy, infertility, tumors	STZ-induced diabetic rats	↑ Serum insulin, ↓ serum glucose, ↓ LDL, ↓ TG, ↓ total cholesterol, ↑ proliferation of β-cells, ↓ oxidative stress	300, 400 mg/kg	84 days	[118,119,120,121]
*Olea europaea* L.	Fruit, pulp, leaves	Diabetes, hypertension, inflammation, diarrhea, respiratory and urinary tract infections, hemorrhoids, rheumatism, laxative, intestinal diseases, asthma, hyper uremia, hyperlipidemia	STZ-induced diabetic rats	↓ α-glucosidase and digestive enzymes activity, ↓ postprandial hyperglycemia, ↑ insulin action, ↑ functionality and survival of β-cells	1mL/100 bw/day (oil)	42 days	[127,128,129,131,163]
*Pinus gerardiana* Wall. ex D. Don	Seeds, leaves, barks	Diabetes, hypertension, sepsis, fungal and microbial infections	STZ-induced diabetic rats	↓ Body weight, ↓ oxidative stress, ↓ hyperglycemia, ↑ expression of PPARγ gene, ↑ Akt, ↑ insulin secretion, ↓ malondialdehyde, ↓ fasting blood glucose levels	3% and 6% *w*/*w*, (Powder)	42 days	[133,135]
*Piper nigrum* L.	Seeds, leaves, flowers, and fruits	Diabetes, menstrual problems, atrophic arthritis, digestive problems, influenza, bacterial infection, inflammation, fever, hypertension, cancer, depressants, diarrhea	Alloxan-induced diabetic rats	↓ ROS generation, ↓ lipid peroxidation, ↓ lipogenesis, ↑ insulin secretion, ↓ blood glucose, ↓ triglyceride, ↓ total cholesterol, ↑ HDL, ↓ LDL, ↑ total antioxidant capacity	50 mg/kg	56 days	[137,141,143]
*Pistacia vera* L.	Seeds, leaves, fruits	Diabetes, coughs, stomach diseases, asthma, sores, chest ailments, rheumatism, trauma, gynecological ailments, hemorrhoids	Pre-diabetic patients	↓ Fasting blood glucose, ↓ insulinemia, ↓ Serum IL-6, ↓ fructosamine, ↓ insulin resistance, ↓ LDL, ↓ malondialdehyde, ↓ proinflammatory cytokines, ↓ glucose absorption	57 g/day	28 days	[145,146,149,164]
*Vitis vinifera* L.	Dried fruits	Diabetes, cancer, obesity, inflammation, hyperlipidemia	T2DM patients	↓ LDL oxidation and LDL-cholesterol, ↓ blood glucose control, ↓ postprandial glucose levels, ↓ HbA1c, ↓ blood pressure	36 g/day	168 days	[152,154,155]
*Zingiber officinale* Roscoe	Roots and rhizomes	Diabetes, digestive disorders, nausea, rheumatism, respiratory tract infection, cough, hypercholesterolemia, neurological diseases, asthma, stroke, constipation, cancer	STZ-induced diabetic rats	↓ Superoxide anion, ↓ hydroxyl radicals, ↓ α-glucosidase and α-amylase activity, ↓cholesterol, ↓ serum glucose, ↓ triglyceride, ↑ HDL, ↑ insulin sensitivity	400 mg/kg	56 days	[158,159,160]

(↓) Decrease; (↑) Increase.

**Table 2 nutrients-15-03266-t002:** Pharmacological effects of the active phytoconstituent/s of the selected antidiabetic plants.

Medicinal Plants	Parts Used	Phytoconstituent Studied	Diabetic Model	Dose Administered	Duration of Treatment	Pharmacological Effects of the Phytoconstituents Used	References
*Althaea officinalis* L.	Leaves, roots, seeds	Lauric acid	Insulin resistance induced in macrophage THP-1 cells	5 μM–50 μM	1 day	Increases glucose uptake in skeletal muscles and improves mitochondrial dysfunction, insulin sensitivity, and GLUT-1 and GLUT-3 expression	[165]
*Anethum graveolens* L.	Leaves, seeds	Carvone	STZ-induced diabetic rats	25, 50, 100 mg/kg	30 days	Alleviates insulin resistance, improves insulin secretion, and reverses glycoprotein abnormalities	[166]
*Allium sativum* L.	Fruits	Allicin	STZ-induced diabetic rats	15, 30, 45 mg/kg	84 days	Improves insulin sensitivity and glucose tolerance, ameliorates diabetes-induced morphological alterations in the kidney, and decreases FBG and triglyceride	[167]
*Brassica oleracea* L.	Leaves	Anthocyanin	T2DM patients	160 mg/kg	168 days	Improves lipid metabolism and insulin resistance, decreases LDL, total cholesterol, and postprandial glucose, and ameliorates diabetic complications	[168]
*Cicer arietinum* L.	Seeds	Quercetin	Alloxan-induced diabetic rats	50 mg/kg	30 days	Decreases blood glucose, total cholesterol, total bilirubin, creatinine, and oxidative stress, regulates glucose homeostasis and improves insulin resistance	[169]
*Cinnamomum verum* J. Presl.	Bark	Cinnamaldehyde	STZ-induced diabetic rats	5, 10, 20 mg/kg	45 days	Elevates HDL level, plasma insulin, and hepatic glycogen and decreases serum glucose, total cholesterol, triglyceride, and LDL level	[170]
*Crocus sativus* L.	Flower stigma	Crocin	T2DM patients	15 mg/kg	84 days	Enhances GLUT-4 expression, inhibits TNF-α, IL-6, alleviates blood glucose, and improves glucose homeostasis and insulin resistance	[171]
*Cuminum cyminum* L.	Seeds	Cuminaldehyde and cuminol	STZ-induced diabetic rats	5, 10 mg/kg	45 days	Increases insulin secretion and insulin sensitivity, lowers blood glucose, provides β-cell protection, and improves lipid profile	[172]
*Eugenia caryophyllata* Thunb.	Flower buds, leaves, stem, fruits	Eugenol	STZ-induced diabetic mice	100 mg/kg bw (I.P. route)	45 days	Lowers blood glucose, blood lipids, and AGEs formation and inhibits α-amylase and α-glucosidase enzymes	[173]
*Foeniculum vulgare* Mill.	Seeds	Kaempferol	STZ-induced diabetic mice	50 mg/kg	84 days	Suppresses gluconeogenesis, enhances glucose uptake in skeletal muscles, and restores hexokinase activity	[174]
*Hordeum vulgare* L.	Grains, leaves, sprouts	β-glucan	STZ-induced diabetic rats	80 mg/kg	28 days	Alleviates diabetic complications, reduces oxidative stress, and lowers blood glucose, total cholesterol, total triglyceride, and LDL level	[175]
*Juglans regia* L.	Husks, kernels, seeds, flowers, bark, leaves	β-carotene	STZ-induced diabetic rats	10, 20 mg/kg	14 days	Improves glucose metabolism and lipid accumulation, lowers inflammatory cytokines, nitric oxide production, and oxidative stress, and enhances glucose uptake in skeletal muscle	[176]
*Lens culinaris* L.	Seeds, sprouts	Saponins	STZ-induced diabetic rats	100, 200 mg/kg	14 days	Ameliorates postprandial hyperglycemia and diabetic complications and inhibits α-glucosidase and aldose reductase enzymes	[177]
*Nigella sativa* L.	seeds	Thymoquinone	STZ-induced diabetic rats	50 mg/kg	28 days	Attenuates blood glucose, lipid peroxidase, nitric oxide production, and oxidative stress and alleviates diabetic nephropathy	[178]
*Olea europaea* L.	Fruits, leaves	Lutein	ARPE-19 cells	0.5–1 μM	24 h	Ameliorates diabetic retinopathy and hyperglycemia, Improves SOD2, HO-1, Nrf2, GSH and catalase regulation	[179]
*Pinus gerardiana* Wall. ex D. Don	Nuts	Linoleic acid	PTPN1, PTPN9, PTPN11 cell lines	0.5–300 μM	7 days	Inhibits the catalytic activity of PTPN1, PTPN9, and PTPN11 and improves glucose uptake by activating AMPK and Akt pathway	[180]
*Piper nigrum* L.	Seeds, flowers fruits, leaves	β-caryophyllene	STZ-induced diabetic rats	200 mg/kg	42 days	Decreases glucose absorption and increases glucose uptake in skeletal muscles, ameliorates glucose tolerance, pancreatic cell damage, oxidative stress, and lipid and blood glucose levels	[181]
*Pistacia vera* L.	Fruits, nuts, leaves	Procyanidins	STZ-induced *db/db* type 2 diabetic mice	250 mg/kg	45 days	Enhances GLUT-4 translocation and glucose uptake on skeletal muscles, possesses insulinotropic ad anti-hyperglycaemic effects	[182]
*Vitis vinifera* L.	Fruits	Ferulic acid	STZ-induced diabetic rats	10 mg/kg	14 days	Alleviates body weight, and blood glucose, attenuates diabetes-associated symptoms, and lowers total triglyceride, total cholesterol, LDL and VLDL levels	[183]
*Zingiber Officinale* Roscoe	Roots	Gingerol	Type 2 diabetic mice (Leprdb/db)	200 mg/kg	28 days	Induces insulin secretion, elevates plasma GLP-1, activates cAMP, PKA, and CREB in the pancreatic islets, and enhances GLUT-4 translocation	[184]

**Table 3 nutrients-15-03266-t003:** Chemical structures of the antidiabetic phytoconstituents of the selected medicinal plants.

Medicinal Plant	Antidiabetic Phytoconstituent	Chemical Structure
*Althaea officinalis* L.	Lauric acid	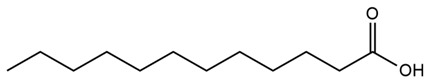
*Anethum graveolens* L.	Carvone	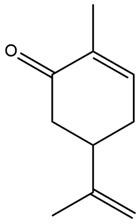
*Allium sativum* L.	Allicin	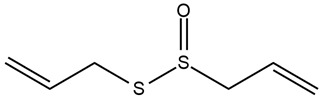
*Brassica oleracea* L.	Anthocyanins	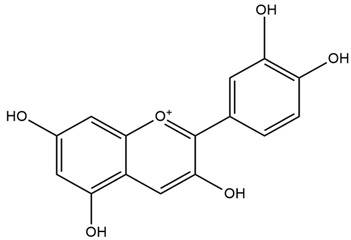
*Cicer arietinum* L.	Quercetin	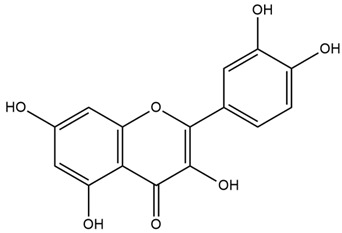
*Cinnamomum verum* J. Presl.	Cinnamaldehyde	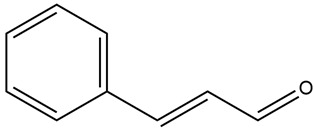 .
*Crocus sativus* L.	Crocin	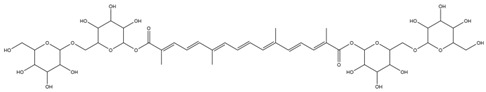
*Cuminum cyminum* L.	Cuminaldehyde	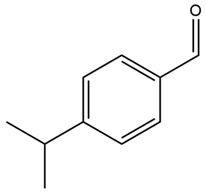
*Eugenia caryophyllata* Thunb.	Eugenol	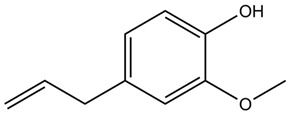
*Foeniculum vulgare* Mill.	Kaempferol	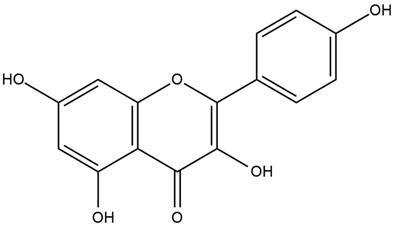
*Hordeum vulgare* L.	β-glucan	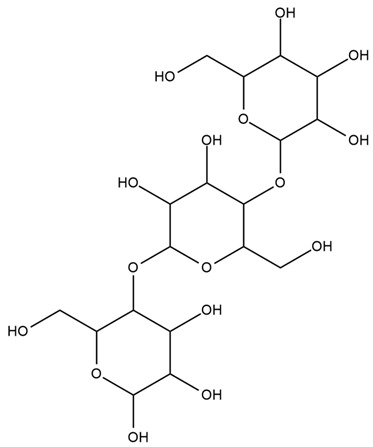 .
*Juglans regia* L.	β-carotene	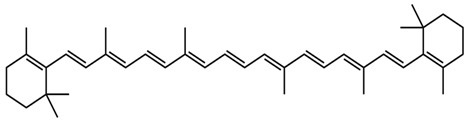
*Lens culinaris* L.	Saponins	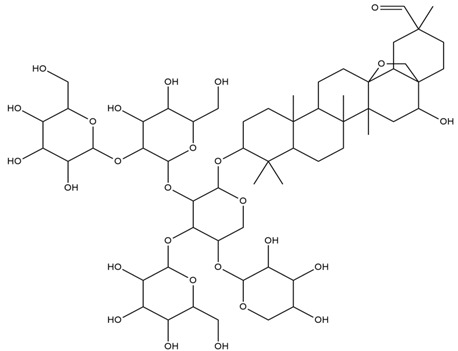
*Nigella sativa* L.	Thymoquinone	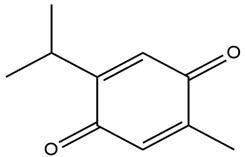
*Olea europaea* L.	Lutein	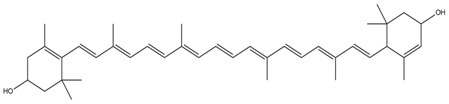
*Pinus gerardiana* Wall. ex D. Don	Linoleic acid	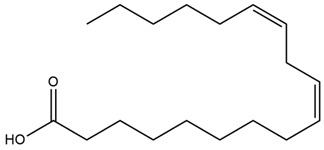
*Piper nigrum* L.	β-caryophyllene	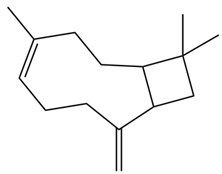 .
*Pistacia vera* L.	Procyanidins	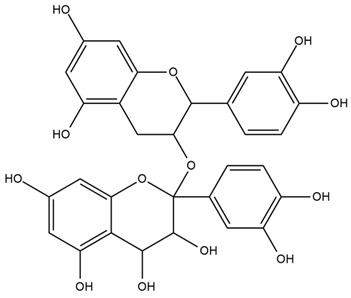
*Vitis vinifera* L.	Ferulic acid	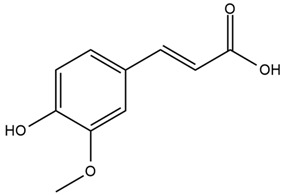
*Zingiber Officinale* Roscoe	Gingerol	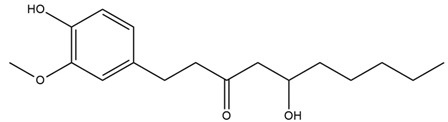

## Data Availability

Not applicable.

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
