# Peer review of "Protective Effects of Medicinal Plant-Based Foods against Diabetes: A Review on Pharmacology, Phytochemistry, and Molecular Mechanisms"

_nutrients, 2023, doi:10.3390/nu15143266_

Round 1

Reviewer 1 Report

Dear Editor, I've read with interest the pa per entitledProtective effects of medicinal plant-based foods against diabetes: A review on pharmacology, phytochemistry and molecular mechanisms” by Prawej Ansari et al. Below my comments-  

General Comment: 

The paper is a well writtes review article on an interesting topic that fully fits the aims of the journal Nutrients. The structure of the review is correct, and the number of references is appropriate. The conclusions are reasonable, and limitations are well stated.

English is good

Reviewer 2 Report

This review article provides a scientific review of medicinal plant-based foods used traditionally for managing diabetes mellitus (DM). It highlights the global prevalence of DM and the limitations of conventional antidiabetic medications due to adverse side effects and limited accessibility, particularly in low and middle-income countries. The review focuses on the therapeutic effects of phytochemicals in medicinal plant-based foods, their molecular mechanisms of action in treating diabetes, and their potential as protective agents. It suggests further research is needed, particularly high-quality clinical trials, to validate the efficacy and safety of these plant-based remedies in DM management. Overall, the review offers scientific insights into the use of medicinal plant-based foods as potential alternatives for DM treatment.

The authors needs to address the comments and suggestions by making necessary modifications to the review to improve clarity, accuracy, and overall quality are given below:

  1. What are the specific phytochemicals in medicinal plant-based foods traditionally used for managing diabetes mellitus (DM), and how do they exert their antidiabetic effects at the molecular level?
  2. What is the clinical evidence supporting the therapeutic effects of medicinal plant-based foods in managing DM, including their impact on lowering blood glucose levels, stimulating insulin secretion, and alleviating diabetic complications?
  3. What is the impact of high-dose and long-term consumption of medicinal plant-based foods on human health, specifically regarding their protective effects against diabetes and potential interactions with conventional antidiabetic medications?
  4. Please modify Table 1 according to the number of days for clarity.
  5. Add DOI in all the references (eg. reference 186, the volume is not mentioned) in the reference section.
  6. It is suggested to mention the importance of synthetic efforts towards development of antidiabetic medicine. In this regard, it is recommended to emphasis the importance of iminosugars and sugar derivatives as an antidiabetic agent .
